# EMERGENT GEOMETRY IN NEURAL REPRESENTATIONS OF THE VISUAL WORLD

## ABSTRACT

How does the brain transform the complex visual world into meaningful representations that facilitate generalization across diverse conditions? One hypothesis is that *the geometric structure of neural manifolds mirrors causal structures in the environment*, facilitating strong generalization across natural contexts. The analysis of neural manifold structure has yielded neuroscientific insights in domains such as navigation and motor control, which often possess simple, low-dimensional structure. Vision, however, presents unique challenges due to its more complex, high-dimensional, hierarchical structure. Leveraging a digital twin model of primate V4 neurons, we conduct targeted *in silico* experiments that allow us to systematically investigate the relationship between the structure of the visual world and its encoding in the visual cortex. Our findings reveal structural equivalences between world properties and neural activity patterns for rotating objects and textures. Specifically, we demonstrate the emergence of equivariant representations that disentangle latent factors such as object identity and orientation. Finally, we demonstrate that these representations facilitate out-of-distribution generalization, as a decoder trained to linearly decode the orientation of one texture can successfully transfer to novel textures. Remarkably, artificial neural networks trained on object recognition tasks exhibit similar geometric principles. These results provide empirical support for the *mirroring hypothesis* in visual processing and suggest universal principles govern the formation of neural representations across biological and artificial vision.

## 1 INTRODUCTION

A fundamental question in biological and artificial vision is how neural systems extract meaningful, generalizable representations from the intricate patterns of light that comprise visual input. Here, we examine whether the geometric structure of neural representations mirrors causal structures in the environment, an idea that we call the *mirroring hypothesis*. Prior work along these lines proposes that this mirroring facilitates generalization across natural contexts (Conant & Ross Ashby, 1970; Yuille & Kersten, 2006).

The mirroring hypothesis is related to the *manifold hypothesis* (Bengio et al., 2013), which posits that real-world data frequently occupy a lower-dimensional manifold embedded in the higher-dimensional data-space. This idea has inspired the analysis of low-dimensional structure within the activity of high-dimensional neural populations—an approach that has been fruitful in making sense of large-scale neural data (Churchland et al., 2012; Cunningham & Yu, 2014; Gao & Ganguli, 2015; Gallego et al., 2017; Chung & Abbott, 2021). It also connects to the concept of *disentanglement* (Higgins et al., 2018), which posits that representations separating independent factors of variation—such as shape and color—facilitate generalization to new contexts. The mirroring hypothesis builds upon these ideas and goes further, proposing that the geometric structure of latent world variables is directly reflected in their neural representation. A concrete example would be a neural system that learns a representation *equivariant* to a transformation such as 2D rotation. In such a system, rotations in the input space would yield corresponding "rotations" in the neural representation space, preserving the circular structure of the transformation and enhancing the system's ability to generalize across different orientations of objects or patterns.

Evidence supporting the mirroring hypothesis can be found across brain regions. For instance, the head-direction circuit in the fruit fly exhibits a ring-shaped geometry in both its synaptic connectivity and functional activity (Figure 1a), directly mirroring the circular nature of head orientation (Wolff

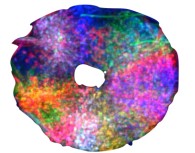
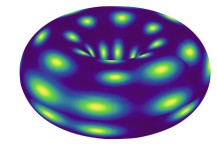
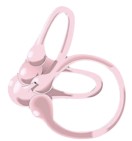

**(A)** Fruit Fly Head-Direction Circuit    **(B)** Grid Cell Firing Pattern    **(C)** Semicircular Canals

**Figure 1: Geometry in the Brain Reflects the Structure of the World**. Neural circuits for (A) tracking head direction Wolff et al. (2015), (B) building spatial maps of the world Gardner et al. (2022); Klindt et al. (2023), and (C) maintaining balance possess rich structure that reflects the geometry of the world. *Credits: (A) Wolff, Iyer, Rubin, (B) Klindt, et al (2023), (C) Science Photo Library.*

et al., 2015; Kim et al., 2017). Similarly, a toroidal topology is observed in the activity of grid cells in the brain's entorhinal circuit for spatial navigation (Figure 1B; Gardner et al., 2022), which reflects the two-dimensional geometry of the surface an animal navigates (with wraparound to avoid singularities). The semicircular canals in the vestibular system provide another notable example (Figure 1C). Situated adjacent to the cochlea, these three orthogonally-oriented, fluid-filled structures in the inner ear detect angular acceleration of the head. Movement of the head causes displacement of the fluid, which is measured by fine hairs lining the canals—and effectively represents head rotation along each of the three orthogonal axes. Remarkably, this anatomical configuration closely approximates the mathematical structure of the special orthogonal group $SO(3)$ of three-dimensional rotations, with each canal corresponding to one axis of rotation. These examples suggest a general computational strategy employed throughout the brain to preserve the geometry of data throughout stages of information processing. This preservation may be a key factor in enabling flexible, efficient information processing, and out-of-distribution generalization (Bernardi et al., 2020), allowing neural systems to leverage the inherent structure of the world in their computations.

The analysis of neural manifold structure has yielded neuroscientific insights in domains such as spatial navigation (Gardner et al., 2022) and motor control (Churchland et al., 2012; Gallego et al., 2017), where the underlying representations tend to be confined to a few key variables. The visual system, however, poses distinct difficulties for such analysis, due to its multi-layered organization, the vast amount of information it processes, and the high-dimensional, hierarchical structure of the input. These complexities make it challenging to map visual neural manifolds across the diverse range of stimulus conditions and variations present in real-world environments.

Here, we tackle these challenges by leveraging "digital twins" of neural populations in the visual cortex. A digital twin is a computational model that accurately predicts the responses of a large population of neurons to arbitrary stimuli (Walker et al., 2019; Bashivan et al., 2019). Digital twins are constructed by recording responses from populations of neurons while the animals observe natural stimuli, and using deep learning to learn the relationship between the stimuli and the resultant neural activity. In this study, we employ a model trained on data from macaque area V4, provided by Willeke et al. (2023), which was collected while the animals viewed natural images (for details of electrophysiological recordings and behavior see Willeke et al. (2023)). The digital twin approach offers two key advantages:

1. **Unlimited experimentation**: Once trained, these models can predict neural responses to virtually any visual stimulus, allowing us to conduct a potentially unlimited number of *in silico* experiments. For example, we can systematically and exhaustively manipulate latent visual features in ways that would be difficult or impossible in physical experiments alone, enabling us to isolate and study specific aspects of visual processing

2. **Precise causal manipulation of neural manifolds**: We can synthesize stimuli to perform precise manipulations in the geometry of the neural manifold, a level of control that is highly difficult to achieve without a differentiable model.

While several prior papers have leveraged point two (Walker et al., 2019; Willeke et al., 2023; Bashivan et al., 2019), in this paper, we primarily take advantage of the first point. To evaluate neural manifolds in these digital twins, we generated two distinct types of datasets: one consisting of image-plane rotations of 3D-rendered objects and another comprising rotations of texture classes with pixel intensities that vary non-smoothly with orientation. This approach allows us to systematically explore how different types of visual transformations are represented in neural activity patterns. Our analysis reveals several key findings:

1. **Structural equivalences between world properties and neural activity patterns**: For rotated images of individual objects, we find that the low-dimensional circular manifolds present in pixel space (mirroring rotational symmetries in the latent space) are preserved within the neural space. This preservation facilitates linear decoding of stimulus orientation from both pixel and neuronal representations.

2. **Emergence of (quasi-) equivariant representations**: Interestingly, for rotating texture classes where the input space lacks an inherent linearly decodable circular structure, we find that such a structure nonetheless emerges within the neural manifold of the digital twin. This emergent structure enables both linear decoding of texture orientation and demonstrates near equivariance, where rotations in the world correspond to equivalent rotations in neural representations.

3. **Disentanglement and cross-condition generalization**: The equivariance in transformations between world and neural representations can enable the disentangling of latent factors, such as texture rotation. This disentanglement facilitates generalization to out-of-distribution combinations of texture and rotation. We demonstrate this through zero-shot transfer of orientation decoders across texture classes.

Remarkably, we find that artificial neural networks (ANNs) trained for object recognition exhibit similar geometric principles to those observed in the V4 digital twins, including structure preservation, equivariance, and cross-condition generalization. These parallels suggest that the principles we uncover may be fundamental to efficient visual processing, transcending the specific architecture of biological or artificial systems. Our work demonstrates that a representational-level analysis of neural manifold geometry can bridge biological and artificial neural systems, uncovering universal principles of visual information processing. By leveraging the power of digital twins, we gain unprecedented insight into the geometric structures that underlie neural representations in the visual cortex, advancing our understanding of both biological and artificial vision systems.

## 2 FRAMEWORK

We begin by first illustrating the problem the brain faces in interpreting the visual world. Importantly, we consider the relationships between structures in the *world*, *data*, and *neural representations*.

**World:** The world contains latent variables with inherent structures—i.e. objects and their poses, sizes, colors, etc. We consider each object and the set of natural transformations that can be performed on it as an *object manifold* (DiCarlo & Cox, 2007; Chung & Abbott, 2021). In Figure 2, we illustrate this point with two objects, whose manifolds are defined by planar rotation.

**Data:** The brain receives information about the world through sensory measurements, such as images. A measurement captures a slice of the world information. In the case of images, the 3D world is projected onto a 2D surface, which often leads to distortions or loss of information in the latent world manifold. Figure 2 illustrates two possible outcomes of this projection:

- *Structure Maintained*: In the top row, the circular structure from the world space is preserved in the image space. This represents cases where the latent structure remains directly observable in the data.

- *Structure Lost*: In the bottom row, the circular structure becomes a scattered point cloud in the image space, representing cases where the projection obscures the underlying latent structure.

**Neural Representation:** The brain transforms sense data into neural representations that facilitate intelligent behavior and inference.

- *When Structure is Maintained*: If the latent variable structure is present directly in the data (i.e., linearly decodable), its reflection in the neural representation is likely expected. As shown in the top row of Figure 2, a (non-degenerate) neural network, trained or untrained, can preserve this structure.

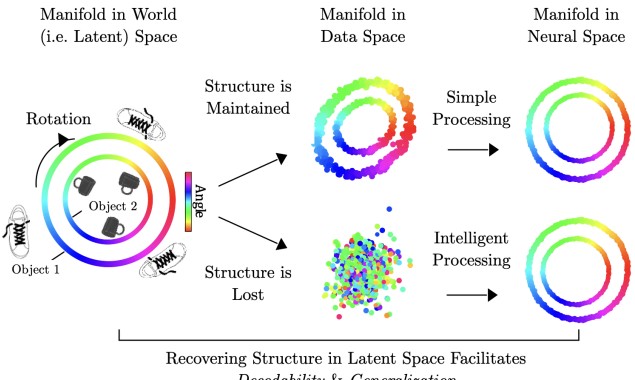

**Figure 2: Graphical illustration of manifold representation across world, image, and neural spaces**: This figure demonstrates the transformation of data manifolds across three spaces. In world (latent) space, rotating objects are represented on a circular manifold with color-coded angles. Shown here are two example manifolds corresponding to a rotating shoe and cup. In the process of taking an image, the latent world structure is projected to a new space, which may distort the world structure. We consider two scenarios: one where structure is maintained, preserving the circular structure, and another where structure is lost, resulting in a scattered point cloud. A deep neural network will tend to preserve structure in the input. However, if the structure is not present linearly in the input, the network will need to learn a nonlinear map that recovers that structure through training.

- *When Structure is Lost*: If the structure is not apparent in the surface-level data, as in the bottom row of Figure 2, according to the *mirroring hypothesis*, the brain must perform non-trivial transformations to recover the latent structure. This scenario is where we observe true information processing: a trained neural network learns to map the scattered points back into a circular structure in neural space.

This framework highlights a critical distinction in neural information processing. When the input data directly reflects the world's structure, the presence of this structure in neural representations may not indicate sophisticated processing. However, when the brain or a trained network recovers structure that is not linearly present in the input, it demonstrates a genuine ability to uncover latent variables and relationships. Understanding this distinction is crucial for interpreting neural manifolds. It allows us to differentiate between cases where neural representations simply relay input characteristics and those where they reveal the system's capacity to extract hidden structure from complex, ambiguous data. This insight is fundamental to understanding how both biological brains and artificial neural networks process information and generalize across diverse contexts. We emphasize, then, the importance of two core questions in the analysis of neural manifold structure (Figure 2):

1. What is the relationship between the structure of the world and the neural manifold? In line with the mirroring hypothesis, does the structure of the neural representation reflect the structure of latent world variables?

2. Is this relationship trivial—i.e. is the structure already linearly present in the data—or does it reflect a sophisticated representational system capable of recovering latent world variables from partial or distorted data?

While question one has been examined to some extend in the literature, question two has received much less attention; the analysis of the structure of the data space is often overlooked. In this paper, we address both questions in the context of neural population responses in the visual cortex, and find that understanding the relationship between *data structure* and *neural representation structure* is critical for drawing meaningful insights about the information processing performed by both biological and artificial visual systems.

## 3 RELATED WORKS

Our work aims to relate manifold structures of three types of variables: causal latent variables in the world, proximal sensory inputs, and neural activity. Many others have hypothesized a close structural relationship between neural activity and the world. In the early 1900s, Gestalt psychologists (Köhler, 1929; Ross, 1949) proposed an isomorphism between physiology and perception, with perceptual structures corresponding closely to structures in the world. Later work in cybernetics proved the

Good Regulator Theorem, which stated that every good regulator of a system must contain a model of that system (Conant & Ross Ashby, 1970). In the context of intelligent behavior, this suggests there should be parallel structure between the brain and the world. Theories that the brain implements a generative model, such as variants of analysis-by-synthesis (Yuille & Kersten, 2006; Dayan et al., 1995) and predictive coding (Rao & Ballard, 1999), also propose that the brain has an internal model of the world. However, they make no claims about geometric correspondences between the representational space and the world it represents, and they do not necessarily propose a relationship as strict as equivariance. In machine learning, a number of studies have proposed architectures that purport to achieve various equivariances as a means of achieving good generalization. These include translation-invariant convolutional networks (LeCun et al., 1989), rotational equivariance (Oyallon & Mallat, 2015), viewpoint equivariance (Hinton et al., 2018), or general group equivariance (Cohen & Welling, 2016). Our work goes beyond these works to predict and examine structure within biological neural systems, such as equivariance, as a strategy for facilitating generalization.

Many studies have used dimensionality reduction techniques to reveal *low-dimensional* structure embedded within high-dimensional neural activity space (e.g. Churchland et al., 2012; Nieh et al., 2021), such that only a few dimensions explain much of the variance within the population activity. Some of this low dimensionality may be caused by limitations in the sensory ensemble (Gao et al., 2017), such that richer, more natural inputs would reveal more dimensions. Nonetheless, neuroscience studies with high-entropy naturalistic movies still find a small fraction of the possible neural space is occupied by neural activity (Stringer et al., 2019).

Another structural aspect of neural manifolds is their *topology*, which can be revealed using techniques like Topological Data Analysis (TDA) based on persistent homology(Gardner et al., 2022; Chaudhuri et al., 2019; Curto, 2017; Hermansen et al., 2022). These techniques identify when two manifolds share similar features—such as the number of holes—allowing them to distinguish between categorically different structures like rings, spheres, and tori. TDA of recordings from hundreds of grid cells in the medial entorhinal cortex demonstrated that the joint activity of these cells resides on a toroidal manifold (Gardner et al., 2022), even when the task is of lower dimensionality such as 1D wheel running (Hermansen et al., 2022).

Finally, *geometric* properties of the neural manifold affect computation and readout, especially linear decodability (DiCarlo & Cox, 2007). Under distribution shift, linear decoding can perform well on cross-conditional generalization (CCG) if the neural dimensions encoding an attribute remain parallel as other attributes are varied, as reported in the hippocampus and prefrontal cortex (Bernardi et al., 2020). A similar concept called manifold capacity uses statistical mechanics to capture the shapes of neural manifolds that allow simple readouts (Chung & Abbott, 2021).

Notably, these various neuroscience and machine learning studies focus on the correspondence between the world and the neural representation, but neglect to address whether this correspondence is trivial because it is present already in the sensory input, or if the relationship is notable because it must emerge through computation. To understand this, we must understand the structural relationships present in the sensory data, which motivates our experiments.

## 4 EXPERIMENTS

We now perform a suite of experiments leveraging digital twin models of the macaque visual cortex, pre-trained image models, and parameterized stimuli to investigate the relationship between geometric structures manifest in the *world*, *data*, and *neural representations*.

### 4.1 DIGITAL TWINS OF THE VISUAL CORTEX

In these studies, we employ a data-driven neural predictive model as a digital twin of the visual cortex to probe the structure of the response manifold of these models when presented with a set of structured input stimuli. Our model architecture, inspired by previous work in the field (Bashivan et al., 2019; Willeke et al., 2023; Cadena et al., 2023), consists of a core shared across neurons and neuron-specific readouts. For the shared core, we utilize a ResNet50 backbone truncated after layer 3, an architecture demonstrated to effectively predict V4 neuronal responses (Cadena et al., 2023). This core is combined with a neuron-specific Gaussian readout mechanism based on (Lurz et al., 2020). The electrophysiological data used in this study, comprising responses from $N = 1244$

single neurons in macaque area V4 to natural images, was generously provided by Willeke et al. (2023). This dataset formed the foundation for our model training and subsequent analyses of neural representations in the visual cortex.

The model processes input as follows: the truncated ResNet50 extracts features, which then undergo batch normalization and ReLU activation. For each of the $N = 1244$ isolated single neurons in macaque area V4, a Gaussian readout learns the neuron's receptive field position during training. This readout extracts a feature vector at the learned position and applies linear regression to map these features to neuronal responses. To comprehensively assess neuronal responses to all test stimuli (objects and textures), we expanded our effective neuron count by replicating each neuron's response across multiple spatial locations. This approach leverages the retinotopic organization of macaque V4 neurons, where individual neurons respond to features at specific receptive field locations. Rather than using a single learned position per neuron, we sampled each neuron's response on a fixed $7 \times 7$ grid of locations that spans the whole image equally. Consequently, for each test stimulus, we obtained a response vector consisting of $N = 1244 \times 7 \times 7$ neuronal responses. We investigate four variants of neural network architectures to compare their manifold structures:

1. **Digital Twin - Trained:** The ResNet50 core (Layer 3) and readout are trained end-to-end on all $N = 1244$ neurons. This model represents our data-driven approach to modeling V4 neuronal responses.
2. **Digital Twin - Untrained:** Identical architecture to (1), but with randomly initialized weights for both the core and readout. This serves as a control to assess the impact of architecture alone.
3. **ImageNet-Trained Core:** The original ResNet50 core (Layer 3) pre-trained on ImageNet classification. This model allows us to compare the neural manifolds of our data-driven V4 model to those of a task-optimized computer vision model. It helps us understand how task-specific optimization shapes the representation space differently from neuron-specific optimization.
4. **Random Core:** The same architecture as (3), but with randomly initialized weights. This serves as an additional control, allowing us to differentiate between the effects of architecture and ImageNet training on the manifold structure.

By comparing the trained V4 neuron-driven digital twin to the trained task-driven image model, we aim to illuminate differences and similarities between the representational strategies adopted by the brain and deep neural networks. Our untrained versions of the digital twin and ImageNet-trained model serve to disentangle the contributions of architecture and training to the model's neural representation structure.

## 4.2 Finding 1: The geometry of the data manifold propagates to neural space

We begin by examining the relationship between the structure of the latent rotation variable present in the world and its structure in the data space. In theoretical analyses, presented in A.5, we demonstrate that the circular structure of an idealized camera axis rotation of an object should be preserved in image (pixel) space when applying Principal Component Analysis (PCA) in an idealized setting. To test this hypothesis empirically, we generated a dataset of objects rotated around a camera axis. We selected 3 objects from the Google scanned objects (GSO) dataset that exhibit natural geometric complexities with only a few symmetries (Figure 3a). Using Kubric (Greff et al., 2022), we rendered scenes in which each object is centrally positioned in a 3D world, a fixed distance above a rectangular plane (floor). The scenes featured the same textured background to focus solely on the object's texture and geometry. The rendering process was standardized across all 3 objects to ensure comparability, using fixed lighting settings and camera angles. Each object was rotated around the axis defined by the center of the camera sensor and the object's center of mass. We considered 180 rotations of each object ranging from $-\pi/2$ (i.e. the object's initial orientation) to $\pi/2$ (Figure 3a).

Our visualizations of PCA on the object datasets (Figure 3b, Left) validate our theoretical results. Despite the complexity of the scenes, which featured textured objects on textured backgrounds (Figure 3a), we observed a low-dimensional, circular manifold even in pixel space. Interestingly, the manifold more closely resembled a saddle shape than a perfect circle. In the appendix, we elaborate on how this saddle shape emerges in the 3D projections of PCs, capturing the periodic nature of sine and cosine functions as they rotate through a full $2\pi$ cycle. When embedding multiple

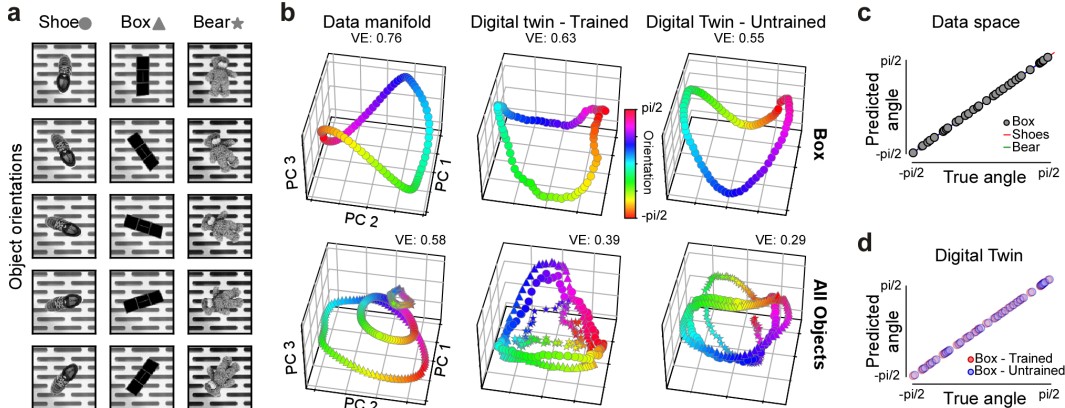

**Figure 3: Visualization of object manifolds across image and neural spaces. a.** Example rotations of three GSO objects (shoe, box, and bear) rendered using Kubric, showing five orientations for each object. **b.** Manifold representations in three different spaces, each visualized using the first three principal components (PCs): Data manifold in image space, trained digital twin's neural space predicting single neuron V4 responses, and untrained digital twin's neural space with random initializations. The top row shows the manifold for a single object (box), while the bottom row displays manifolds for all three objects simultaneously. Each point represents an object orientation, color-coded by rotation angle from $-\pi/2$ to $\pi/2$. **c.** Decoding orientation is possible both in image space and neural space. Comparison of predicted versus true angles for all three datasets (Shoes, Box, and Bear) across different models. The top panel shows the raw data, the middle panel displays results in trained digital twin space, and the bottom panel illustrates the decoding performance for an untrained digital twin. Each point represents a single prediction, with the x-axis showing the true angle and the y-axis showing the predicted angle. Each dataset was split into training (80%) and testing sets (20%). The diagonal line represents perfect prediction.

objects into the same 3D PC space, we found that object manifolds were stacked along the third axis. This suggests that while the first two components primarily encode rotational information, the third captures object-specific features.

Next, we examine whether this structure is preserved in both trained and untrained digital twins. In line with expectations, Figure 3 illustrates that the structure is preserved. The preservation of manifold structure across spaces suggests that object rotation information should be linearly decodable from each representation. Our results confirmed this: we successfully decoded object angles using linear regression in all three spaces with comparable, near perfect test accuracies and mean (across objects) angular errors of 0.007 radians, 0.008 radians, and 0.011 radians for pixel, trained, and untrained spaces, respectively (Figure 3c – see appendix A.2 for more details on linear regression application). Notably, even randomly initialized networks preserved this structure, enabling angle decoding. This aligns with prior research on input structure preservation in neural networks Poole et al. (2016), extending such findings to complex, real-world visual stimuli.

### 4.3 FINDING 2: THE VISUAL CORTEX RECOVERS THE WORLD MANIFOLD FROM AN UNSTRUCTURED DATA MANIFOLD

In the previous analysis, we consider a case in which the latent world structure is directly preserved in the data. We now consider a case in which latent variables are not preserved (linearly decodable) in the image space. To investigate this, we created a dataset of rotating textures where each texture class maintained consistent statistics across orientations but showed no systematic pixel-space relationships (Figure 4a). We developed two texture categories: "synthetic" and "naturalistic". For synthetic textures, we randomly placed bars or arrows in each image for every rotation. For naturalistic textures, we used the parametric texture model by (Portilla & Simoncelli, 2000), identifying oriented textures from the DTD dataset (Cimpoi et al., 2014) via an oriented Gabor filterbank (see appendix A.1). We randomly selected two textures with strong orientable features, termed "Banded" and "Stratified", then generated texture metamers for each orientation, ensuring matching texture statistics but low pixel-level correlations. It is worth noting that these texture datasets represent just one example where image space lacks structure. Many other scenarios, such as object rotation with occlusion, likely present similar challenges to visual processing: in fact, most natural visual scenarios involve such complexities.

Visualization of the image manifold for these rotating texture classes revealed no circular low-dimensional structure in PC space (Figure 4b). In addition, we find that texture orientation cannot be linearly decoded from pixel space (mean angular error across textures of 0.6 radians) (Figure 4c). However, processing through a digital twin recovered the world manifold, yielding a circular low-dimensional structure that enabled linear decoding of texture orientation (mean angular error across textures of 0.12 radians). An untrained network, in contrast, lacked a clearly structured low-dimensional manifold and exhibited lower decoding performance (mean angular error across textures of 0.24 radians) than the trained network, demonstrating that the architecture alone was not sufficient to explain the high decoding accuracy. Nevertheless, the fact that the decoding accuracy of the untrained network was still higher than in pixel space indicates that the neural network architecture did play a role.

Decoding performance in the trained model generally increased across layers, peaking near the read-out layer, despite some fluctuations along the way (Figure 4d). In contrast, the untrained network showed an overall increase in performance across layers initially, but this trend reversed and declined toward the later layers, with fluctuations throughout. These effects further underscore the crucial role of training in learning representations that recover world structure.

Our findings suggest that the trained neural network may exhibit near-*equivariance* with respect to texture rotations. Equivariance is a concept from group theory that has proven useful in constructing artificial neural networks that respect the geometric structure of the data they represent (Cohen et al., 2021). Groups are algebraic descriptions of sets of transformations that meet certain axioms. Many of the transformations that structure the visual world are describable as groups—including translation, rotation, and scaling. Intuitively, equivariance implies that a transformation applied to input textures results in a corresponding transformation within the network's internal representations, potentially enabling generalization across various transformations of the same texture. Formally, we say that a function $\phi : X \mapsto Y$ is equivariant to a group $G$ if $\phi(gx) = g'\phi(x)$ for all $g \in G$ and $x \in X$, with $g' \in G'$, a group homomorphic to $G$ that acts on the output space. Concretely, a rotation of an image would result in a corresponding (homomorphic) "rotation" in the representation space.

To test this hypothesis, we fit a 2D rotation matrix to the first 2 principal components of the activations from the trained and untrained networks for our four textures utilizing a projected gradient scheme described in A.4 . As seen in figure 4e, a rotation matrix fit to the trained network's output exhibited markedly superior predictive performance. This suggests that training enhances the network's equivariance with respect to texture rotations, enabling it to learn a more systematic internal representation of orientation. These findings demonstrate the brain's capacity to recover latent world structure from seemingly unstructured visual input. Moreover, we demonstrate emergent near-equivariance in neural representations in the visual cortex—a novel finding that suggests that equivariance may be a fundamental principle of representations in the brain.

## 4.4 FINDING 3: EQUIVARIANT NEURAL REPRESENTATIONS FACILITATE GENERALIZATION

The structure of neural representations plays a crucial role in a system's capacity to generalize. Manifold alignment—where similar features across different stimulus classes are represented in consistent ways—may be a key mechanism underlying this ability. For example, in textures with similar symmetries, such as elongated textures, when their elongated axes are aligned (e.g., vertically), their neural representations also become aligned in neural space. Such alignment could allow a decoder trained to decode orientation from one texture to transfer directly, without additional training, to other textures whose neural manifolds are similarly aligned. When visualizing the manifolds of two texture classes in the same low-dimensional neural space, we observed that the trained network aligned texture orientations across classes, positioning similar orientations in close proximity (Figure 5a). This alignment was absent in the untrained network. We hypothesized that such manifold alignment might be advantageous for generalization.

To test this hypothesis, we evaluated cross-condition generalization (CCG) (Bernardi et al., 2020). Namely, we trained a decoder to determine the angle of one texture class and then evaluated its performance on other texture classes (Figure 5b). This process was repeated for all possible combinations. The trained model exhibited strong generalization capabilities (Figure 5c): decoding performance for texture classes not used in training was only marginally lower than for the training class (avg. diagonal circular correlation: 0.96 - avg. off-diagonal circular correlation: 0.83). Interestingly, the

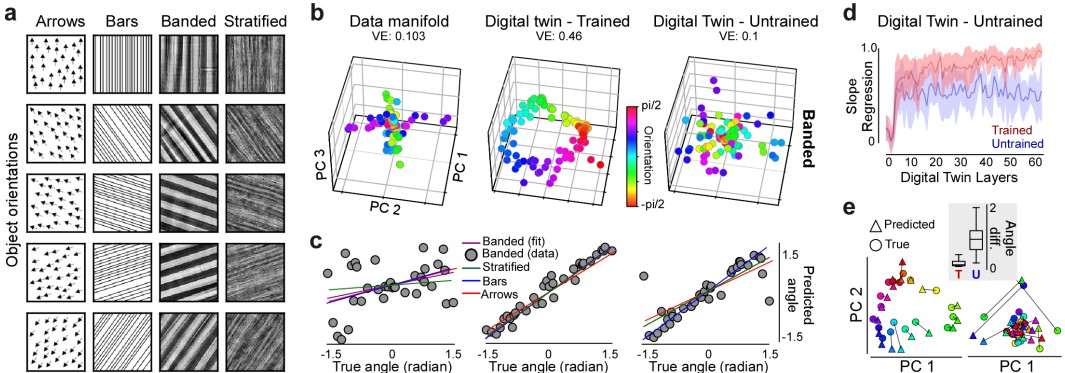

**Figure 4: Neural network trained on V4 data recovers latent structure from unstructured image data.**
**a.** Example rotations of four texture classes (Arrows, Bars, Banded, Stratified), showing five orientations for each texture type. **b.** Manifold representations of an example texture in three different spaces, visualized using the first three principal components (PCs): Data manifold in image space, trained digital twin's neural space, and untrained digital twin's neural space. Each point represents a texture orientation, color-coded by rotation angle from -π/2 to π/2. "VE" label indicates Variance Explained by the first three PCs. **c.** Decoding of texture orientation using regression. Scatter plots show the relationship between true angles and predicted angles for the data manifold, trained digital twin, and untrained digital twin. **d.** Decoding performance of orientation across model layers for both untrained and trained digital twins. The plots show mean slope of the fitted lines between true and predicted angles of our four textures across different layers of the models. The shading denotes the standard deviation. **e.** A rotation matrix is fit on 20 random 80-20 splits of the first 2 principal components of the (un)trained digital twin output on the textures using a scheme described in A.4. The test loss (Average Angular Displacement (rads)) in all 80 trials is recorded and plotted as a box plot. Also, a rotation matrix fit on all the banded (un)trained network output has a subset of its predictions shown compared to its true targets. In either case, the matrix fit to the trained output performs better.

untrained network showed some capacity to decode angles for its training class (Figure 5c) (avg. diagonal circular correlation: 0.88), indicating that texture angle information is preserved in the high-dimensional neural space. However, decoders trained on untrained model representations failed to generalize across other classes (Figure 5c) (avg. off-diagonal circular correlation: 0.39), suggesting suboptimal alignment of manifolds across textures.

To investigate whether this manifold alignment is specific to the visual cortex or a more universal property of intelligent vision systems, we extended our analysis to a robust ResNet model trained for image classification (Figure 5d-f). We specifically examined layer 3 of this network, as previous work has shown that this layer corresponds well to macaque V4 Willeke et al. (2023), providing a fair comparison to our DDM. Remarkably, the results from the robust ResNet closely mirrored our findings from the brain-trained model. The ResNet recovered the low-dimensional structure present in the latent space but not in the image space, which allowed for successful linear decoding of texture angle. Decoders trained on one texture class generalized well across other classes, similar to the DDM. In contrast, a randomly initialized network with the same architecture failed to show this cross-class generalization.

These findings suggest that the ability to align manifolds and generalize across different texture classes is not unique to the models of the brain but may be a common feature of trained intelligent vision systems. This alignment facilitates generalization by creating a consistent representational structure across varied inputs. In contrast, untrained networks, while capable of capturing some angle information, lack the organized representation necessary for effective cross-class generalization.

## 5 DISCUSSION

Our findings provide compelling evidence for the mirroring hypothesis in visual processing, revealing a principle underlying both biological and artificial vision systems. We demonstrate that neural representations in area V4 of the primate visual cortex, as well as in trained artificial neural networks, reflect geometric structures of latent variables in the visual world. We further demonstrate that the visual cortex is able to recover and reflect the structure of these latent variables even when they are not linearly decodable in the input space. Our results reveal the emergence of near-equivariant

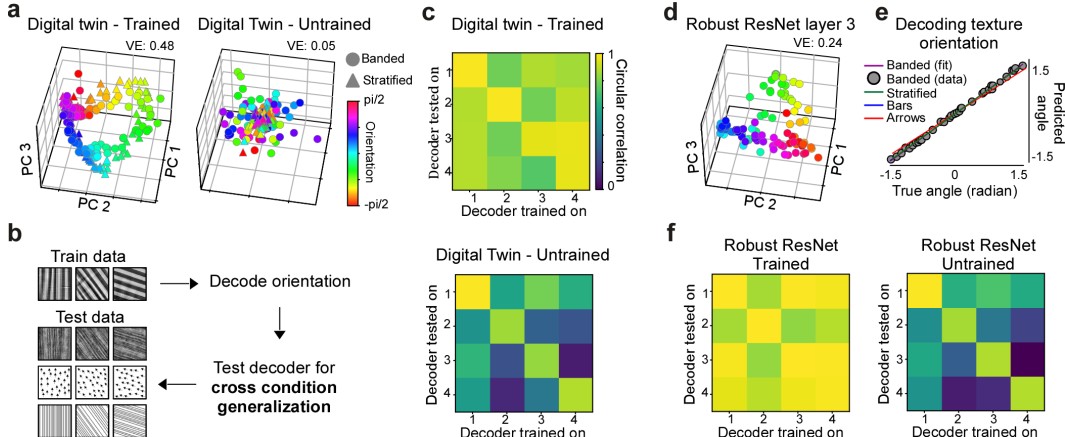

**Figure 5: a.** Manifold representations of two texture types embedded in the same space for trained and untrained digital twin neural spaces, visualized using the first three principal components (PCs). Points are color-coded by orientation angle from $-\pi/2$ to $\pi/2$. The trained model shows alignment of the manifolds for different textures, while the untrained model shows less structured representation. **b.** Schematic of the cross-condition generalization experiment. Decoders are trained on one set of textures and tested on different textures to assess generalization of orientation representation. **c.** Cross-condition generalization performance for the trained and untrained digital twin, shown as a matrix of circular correlations between true and predicted angles. **d.** Manifold representation of texture orientations in layer 3 of a robust ResNet, visualized using the first three PCs. **e.** Decoding performance of ResNet core on texture orientation, shown as a scatter plot of true vs. predicted angles. **f.** Cross-condition generalization performance matrices for the untrained digital twin, trained robust ResNet, and untrained robust ResNet.

representations in the visual cortex, a property that allows the visual system to maintain consistent relationships between transformations in the input and their neural representations. This equivariance, we argue, is key to the system's ability to generalize across diverse conditions and disentangle independent factors of variation in visual scenes. These findings not only advance our understanding of biological vision but also offer critical insights for the development of more robust and adaptable artificial vision systems, potentially bridging the gap between natural and artificial intelligence in visual processing.

While our studies reveals cases where neural representations closely mirror the structure of the visual world, it's important to note that the brain often reshapes world structures in non-trivial ways. Consider, for example, color perception. In the physical world, visible light exists on a linear spectrum of frequencies. However, our visual system wraps the ends of this spectrum, connecting the longest visible wavelengths (red) with the shortest (violet) to form a perceptual color circle (Shepard, 1962). Another notable example is musical pitch perception. Similarly, while sound frequency increases linearly, our auditory system perceives octaves as having a certain equivalence, effectively wrapping the linear frequency space into a helical structure (Shepard, 1964). Understanding the function of these neural warpings will be critical for developing a comprehensive theory of neural information processing.

Our work demonstrates that the relationship between world structure and neural representations is both fundamental and nuanced. While the visual system often mirrors the latent structure of the world, as we've shown in V4, it can also reshape this structure in ways that may facilitate perception and cognition, as seen in color and pitch perception. Future research should investigate how these principles of mirroring and reshaping extend to other latent features in more complex richer visual scenes and should be studied across the visual hierarchy, as well as other sensory modalities and cognitive domains. By continuing to unravel these principles, we move closer to a unified theory of information processing that spans both biological and artificial intelligence. Ultimately, this line of research opens up a new avenue for exploring how the brain—and artificial systems modeled after it—encode, process, and affect the world around us.

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

# A    APPENDIX

## A.1    SELECTION OF DTD TEXTURES VIA GABOR KERNEL FILTERING

We analyzed texture orientations in the Describable textures dataset (DTD) using Gabor filters, which are particularly effective at detecting directional features. The process involves applying a series of 16 Gabor filters, each tuned to a specific orientation, to grayscale versions of the input images. To quantify the degree of orientation in each texture, we calculated a "dominance ratio", defined as the inverse of the mean Gabor filter response across all orientations. This ratio effectively distinguishes between strongly oriented textures, which produce high ratios due to significant response variations across filter orientations, and more isotropic textures, which yield lower ratios due to more uniform filter responses. The algorithm classifies textures with ratios exceeding a chosen threshold of 6 as having significant directional patterns. By ranking textures based on this ratio, the method allows for a selection of the most strongly oriented textures within the dataset. This approach provides a simple quantitative measure for texture analysis.

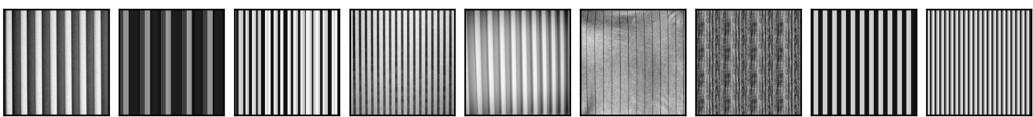

**Figure 6:** An example sample of 9 texture images from the DTD dataset displaying strong vertical orientation.

## A.2    LINEAR REGRESSION FOR ORIENTATION DECODING

Throughout this study, we investigated orientation decoding using two approaches: we either attempted to decode the angle of rotation from neural network responses or directly from pixel space. The circular nature of angular data necessitated special consideration in our methodology. To address this, we converted angles from degrees to radians and decomposed them into their sine and cosine components. We implemented two independent linear regression models: one for predicting the cosine and another for predicting the sine of the target angle. The final angle prediction was derived by applying the arctangent function (atan2) to the predicted sine and cosine values, yielding results in radians. We then evaluated the accuracy of these predictions using circular correlation, a metric specifically designed to account for the periodic nature of angular data. This approach provided a robust measure of the alignment between predicted and actual rotation angles, enabling a more accurate assessment of our orientation decoding techniques in both neural network response space and pixel space.

## A.3    SOME DETAILS ON CROSS-CONDITION GENERALIZATION

To investigate the cross-condition generalization (CCG) of our rotating object and texture datasets, we first normalized each dataset with respect to the statistics of ImageNet (for neural network processing) and then trained Linear Regression models on each of the model output datasets. In the case of rotating objects, our data are rendered to be statistically similar to naturalistic stimuli. However, this is not true for 2 of our artificially generated textures, namely those of oriented arrows and bars. We therefore allowed the batch normalization layers of the trained network to adjust in train mode due to the out-of-distribution nature of the two textures (Fig5c, f). Linear regressions were then trained to predict the rotation angles based on network responses, or pixel intensities. Our results show that the trained models generalize well both within and across texture types, especially in train mode, while the untrained network performs slightly worse (Fig5c, f), with poor cross-generalization in train mode but some improvement when in evaluation mode (Fig7).

We argue that the trained network in train mode adapts effectively to out-of-distribution data via batch normalization, enabling better cross-generalization and we believe exploring generalization under adapting batch normalization layers merits an exploration of its own (Schneider et al. (2020)). In contrast, deviating from batch normalization initialization in the untrained network prevents it from cross-generalizing, possibly due to the effectiveness of its initial batch normalization statistics. Therefore, for a fair comparison, we recommend showing trained network results in train mode and untrained network results in evaluation mode. Figures for the remaining cases are shown below and

display CCG matrices across pixel space, trained space, and untrained space for the four textures and three additional datasets of rotating objects. All CCG matrices are averages over 20 runs.

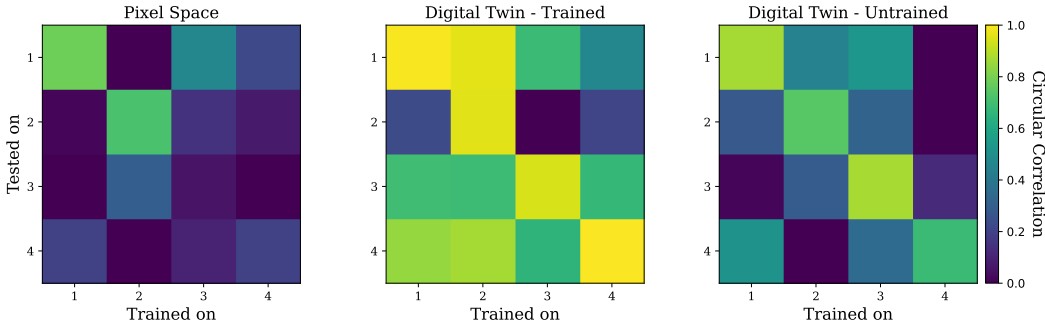

**Figure 7:** Texture cross-condition generalization matrices computed in pixel space (left), trained digital twin (middle) and untrained digital twin (right) neural spaces. The trained and untrained CCG matrices are computed in evaluation mode (the batch normalization statistics are those acquired during training on naturalistic stimuli) and training mode, respectively. First, we notice that texture orientation cannot be decoded in pixel space. Then the trained digital twin has considerable cross-generalization power, especially for rows 3 & 4 (banded and stratified textures), while the untrained network seems to predict its own test sets but mostly fail to cross-generalize.

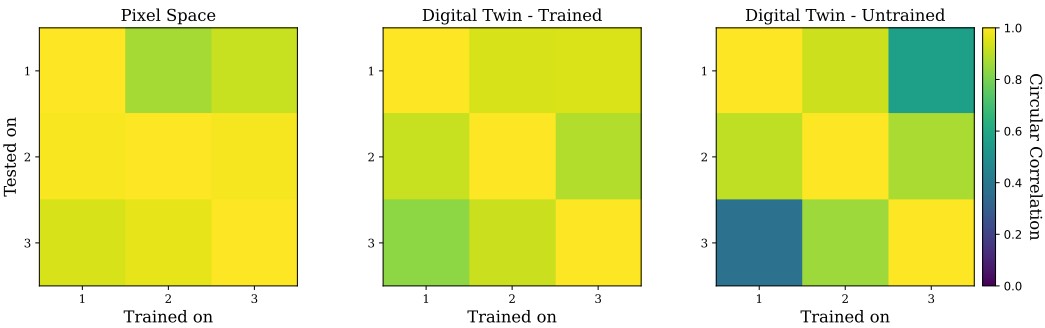

**Figure 8:** Object cross-condition generalization matrices computed in pixel space (left), trained digital twin (middle) and untrained digital twin (right) neural spaces.

### A.4 FITTING ROTATION MATRIX TO DATA

We utilize a scheme from Wen & Yin (2010) for optimizing a matrix over the space of orthogonal matrices. Applied to our setting, let $X$ be a matrix in $\mathbb{R}^{2 \times N}$, where $N$ denotes the number of data points, and each column represents the first two principal components. We assume the columns are organized by orientation of their respective images. We choose our loss function to be given by

$$F(R) = \|RX_{-1} - X_1\|_2^2$$

where $X_{-1}$ is our data matrix without the last column and $X_1$ is our data matrix without the first column. We begin by initializing a random 2D rotation matrix $R$. The gradient is then computed as $G := \partial F(R)/\partial R_{ij}$ and the updated $R$ is given by

$$\left(I + \frac{\tau}{2}A\right)^{-1}\left(I - \frac{\tau}{2}A\right)$$

with $\tau$ representing the learning rate and $A$ defined as $A := P_X G X^T - X^T P_X^T$ where $P_X$ is given as $P_X = (I - XX^T/2)$. Notably, this projected gradient scheme ensures that at each step of optimization we have an orthogonal matrix. Further details on the derivation of this scheme can be found in Wen & Yin (2010). For each of our fits, we use 1000 gradient steps with a $\tau$ of $10^{-5}$. In figure 9 we have completed the exact experiment as in figure 4e except we have now completed the

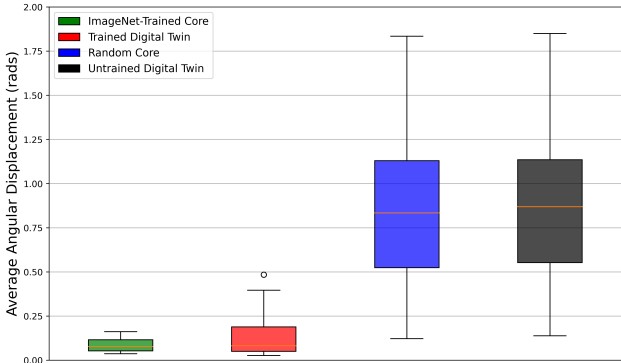

**Figure 9:** A rotation matrix is fit on 20 random 80% subsets of the first 2 principal components of the (un)trained neural output of the digital twin and ImageNet-Trained Core over our four textures. The test loss (Average Angular Displacement) between predictions and targets in all 80 trials is recorded and plotted as a box plot. The matrices trained on the trained outputs performs significantly better.

analysis for the ImageNet - Trained Core and the Random Core for comparison. The rotation matrix fit to the trained core performs significantly well, much better than its random counterpart, suggesting that an artifical network can also learn to be approximately equivariant with respect to orientation.

### A.5 ROTATION OF IMAGE LEADS TO EMERGENCE OF STRUCTURE IN PIXEL SPACE

As seen in Figure 3, plotting the first 3 principal components of our manifold of rotations leads to a "saddle"-like structure in pixel space. A natural question is whether this phenomenon is something unique to our chosen images, or is instead more general. Here, we provide an argument in an idealized setting that could shed some light on this question.

Let us consider a circular region in a square image, located at the center of the frame. Let us also assume that every pixel in this circular region is going to be rotated by $\theta \in [0, 2\pi]$ in $n$ equiangular steps about its center. We define a polar pixelation of our object, with $n$ pixels per ring, and an arbitrary $k$ rings (polar bands). We assume each pixel takes on some real value.

As each step of this rotation is conducted, the set of intensities in each ring remains unchanged, only being cyclically shifted forward by one step, until it eventually loops back to its starting position. Let's call our data matrix $X \in \mathbb{R}^{n \times kn}$, were each row corresponds to the set of all the pixels at one step in the rotation, and each column corresponds to the values of a given pixel across all rotations. Before conducting PCA on this matrix, we will center the columns of $X$.

Note that PCA is invariant to permuting the rows and columns of a data matrix. Namely, we can reorganize the rows to guarantee that each subsequent row corresponds to the data from the next step of rotation. Furthermore, we can organize our data matrix in $k$ blocks, where each block corresponds to the pixels on a given ring. This formulation yields a matrix $X$ that is of the form

$$X := [\ C_1 \mid C_2 \mid \dots \mid C_k\ ] = \begin{bmatrix} c_1^1 & \dots & c_n^1 & c_1^2 & \dots & c_n^2 & \dots & c_1^k & \dots & c_n^k \\ \vdots & \ddots & \vdots & \vdots & \ddots & \vdots & \dots & \vdots & \ddots & \vdots \\ c_2^1 & \dots & c_1^1 & c_2^2 & \dots & c_1^2 & \dots & c_2^k & \dots & c_1^k \end{bmatrix} \quad (1)$$

where each $C_i$ is a circulant matrix and $c_i^j$ is the $i^{th}$ element in the $j^{th}$ circulant matrix.

**Lemma 1.** *Let $X \in \mathbb{R}^{n \times kn}$ be a block matrix $X \in \mathbb{R}^{n \times kn}$ as in equation 1, with centered columns and $n > 1$. Each principal component $\vec{p}$ (corresponding to nonzero explained variance) can be expressed as $E\lambda_i \vec{v_i}$, where $E$ is some fixed constant, and $\lambda_i$, $\vec{v_i}$ are an eigenvalue - eigenvector pair of some symmetric, column centered circulant matrix $C \in \mathbb{R}^{n \times n}$, with $\lambda_i > 0$.*

*Proof.* The typical first step in conducting PCA on a centered matrix $X$ is to compute the empirical covariance matrix $\Sigma := X^T X / (n-1)$. Alternatively, one can consider the eigenvector - eigenvalue

equation of

$$\frac{1}{n-1}XX^T\vec{v} = \lambda\vec{v}$$

and consider the case when $\lambda > 0$. Indeed, multiplying by $X^T$ on each side gives

$$\left(\frac{1}{n-1}X^TX\right)\left(X^T\vec{v}\right) = \lambda\left(X^T\vec{v}\right)$$

Note that since $\lambda > 0$, we have that $X^T\vec{v} \neq 0$. More explicitly, multiplying by $v^T$ in the original equation yields $\|X^T\vec{v}\|_2^2 = (n-1)\lambda\|\vec{v}\|_2^2 > 0$. Hence, if we find an eigenvector - eigenvalue pair of $\frac{1}{n-1}XX^T$, $(\vec{v}, \lambda)$, we can find an eigenvector - eigenvalue pair of $\frac{1}{n-1}X^TX$ as $(X^T\vec{v}, \lambda)$.

Now, let's demonstrate that we can generate any given eigenvector of $X^TX$ with positive eigenvalue in this manner. Let us consider the eigenvector - eigenvalue equation of $\frac{1}{n-1}X^TX$, namely

$$\frac{1}{n-1}X^TX\vec{w} = \mu\vec{w}$$

or

$$\left(\frac{1}{n-1}XX^T\right)(X\vec{w}) = \mu(X\vec{w}),$$

where we have multiplied by $X$. Using the same argument as before, any eigenvector - eigenvalue pair $(\vec{w}, \mu)$ of $\frac{1}{n-1}X^TX$, for $\mu > 0$, can be converted to a eigenvector - eigenvalue pair of $\frac{1}{n-1}XX^T$ as $(X\vec{w}, \mu)$. Now, we could convert this back to an eigenvector of $\frac{1}{n-1}XX^T$ in the form of $(X^TX\vec{v}, \mu)$. But $\vec{v}$ is an eigenvector of $X^TX\vec{v}$, so this is really our intial eigenvector - eigenvalue pair $(\vec{v}, \mu)$. Hence, we will not fail to generate any eigenvectors of $X^TX$ with nonzero eigenvalue using this method. Moving on, we see that

$$XX^T = \sum_{i=1}^{k}C_iC_i^T := C.$$

Moreover, the product, sum, and transpose of a circulant matrix is also circulant, so $C$ is circulant. Furthermore, $C$ is clearly symmetric. Additionally, since each $C_i$ has centered columns, it is not too difficult to see that $C$ also has centered columns.

Let $V \in \mathbb{R}^{n \times n}$ be defined as the column stacking of all the normalized eigenvectors of $C$. Hence, to convert this $V$ to consist of eigenvectors of $\frac{1}{n-1}X^TX$, we compute $\frac{1}{n-1}X^TV$. Finally, to compute our principal components corresponding to nonzero eigenvalues (or nonzero explained variance), we cut out the columns of $V$ corresponding to eigenvalues of 0, to make $V'$. Then, we project our initial data onto $\frac{1}{n-1}X^TV'$ by computing $\frac{1}{n-1}XX^TV'$. Note that $V'$ has columns which are eigenvectors of $XX^T$. Therefore,

$$\frac{1}{n-1}XX^TV' = \frac{1}{n-1}\begin{bmatrix} | & | & \cdots \\ \lambda_1\vec{v}_1 & \lambda_2\vec{v}_2 & \cdots \\ | & | & \cdots \end{bmatrix}$$

Hence, each principal component $\vec{p}$ of our data matrix $X$ that corresponds to nonzero explained variance can indeed be expressed as $E\lambda_i\vec{v}_i$, where $E$ is the fixed constant $\frac{1}{n-1}$, and $\lambda_i, \vec{v}_i$ are the eigenvalue - eigenvector pair of some symmetric, column centered circulant matrix $C$, with $\lambda_i > 0$.

$\square$

One eigenvector of all circulant matrices is a vector of all ones, $\vec{1}$, with eigenvalue equal to the sum of any row or column of the given matrix. However, the matrix $C$ from Lemma 1 has centered columns. Therefore, this particular eigenvector will have eigenvalue 0, and hence will not supply nonzero explained variance in PCA. Furthermore, since our matrix is circulant and symmetric, one real formulation of its remaining eigenvectors is given by $\{\cos(2\pi dj/n) : j = 0, \ldots, (n-1)\}$ and $\{\sin(2\pi dj/n) : j = 0, \ldots, (n-1)\}$ for fixed $d \in \{1, \ldots n-1\}$Park (2002). In other words, the remaining eigenvectors can be chosen to look like discretized versions of sines and cosines. Notably, each one of these "cosine" and "sine" eigenvectors share the same respective value of $d$, they will

also have precisely the same eigenvalue (and norm)Park (2002) . Hence, we can divide our remaining eigenvectors up into pairs of discretized cosines and sines that share the same frequency. The only exception is when $n$ is even, in which case there will be only one eigenvector corresponding to $j = \lceil \frac{n-1}{2} \rceil$, namely $[1, -1, 1, \ldots]$. However, this eigenvector would only have a large eigenvalue if our initial image was dominated by an incredibly high frequency, which is not typically the case for any image that is remotely natural. Hence, we will assume this eigenvector will virtually never be among the first few eigenvectors with the largest eigenvalue. Furthermore, we will make the simplyfing assumption that only sine and cosine vectors of the same frequency share the same eigenvalue, which should be very likely for images that exhibit enough complexity.

Therefore, if were to perform PCA on our matrix $X$ to extract its first three principal components, our first two components will likely look like some discretized sine and cosine of the same frequency, scaled by the same constant. Hence, forming a scatter plot of just these principal components will generate a set of equiangular points on a circle. If we also plot the third principal component, it will also act like some separate sinusoidal, and will generate a "wiggle" on top of our circle, revealing the desired "saddle"-like structure. Notably, for natural images, lower frequencies will tend to be more prevalent - this could give some explanation why we see a small number of "humps" in our circular structure in Figure 3.

We proceed by analyzing a practical scenario where we begin with three functions representing Gabor filters rotating through phase shifts:

$$\mathbf{f}_1(x) = \sin(x), \quad \mathbf{f}_2(x) = \sin(x + \frac{\pi}{4}), \quad \mathbf{f}_3(x) = \sin(x + \frac{\pi}{2})$$

These three functions form the basis for the following analysis. The functions $\mathbf{f}_1(x)$, $\mathbf{f}_2(x)$, and $\mathbf{f}_3(x)$ are equivalent to:

$$\mathbf{f}_1(x) = \sin(x), \quad \mathbf{f}_2(x) = \sin(x) + \cos(x), \quad \mathbf{f}_3(x) = \cos(x)$$

up to a scaling factor. By performing PCA (Principal Component Analysis) on these functions, we extract basis functions that describe the underlying structure of these signals.

**Phase Shifted Gabor Functions:** Figure 10 shows the three Gabor functions $\mathbf{f}_1(x)$, $\mathbf{f}_2(x)$, and $\mathbf{f}_3(x)$ corresponding to the sine, phase-shifted sine, and cosine functions respectively. These functions represent different rotations and combinations of sine and cosine terms, which will later form the basis for our analysis using PCA.

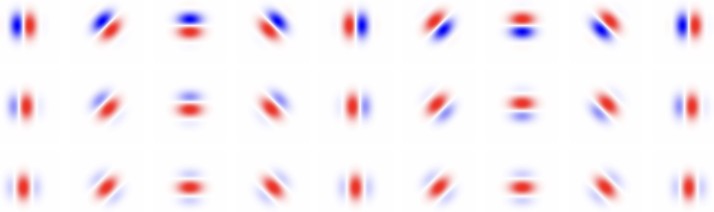

**Figure 10:** Rotating Gabor functions: $\mathbf{f}_1(x) = \sin(x)$, $\mathbf{f}_2(x) = \sin(x + \frac{\pi}{4})$, $\mathbf{f}_3(x) = \sin(x + \frac{\pi}{2})$. The plots demonstrate how the Gabor filters rotate with respect to phase shifts.

**Principal Component Analysis (PCA):** Performing PCA on these functions reveals interesting properties of the resulting basis functions. In Figure 11, the rows represent the ordered basis functions obtained from PCA for each corresponding function. The first row corresponds to $\mathbf{f}_1(x)$, the second to $\mathbf{f}_2(x)$, and the third to $\mathbf{f}_3(x)$.

**Observations:** Based on these results, we make three observations. Firstly, *DC Component*: The second and third functions ($\mathbf{f}_2(x)$ and $\mathbf{f}_3(x)$) contain a DC component, which is the mean or constant offset in the signal. Secondly, *Quadrature Pairs*: After the DC component is subtracted, the remaining components emerge in quadrature pairs. These pairs resemble harmonics, such as the odd-even "Pepsi sign" (cosine-sine pairs). Thirdly, *Odd and Even Frequencies*:



**Figure 11:** Basis functions obtained from PCA for the three Gabor functions. Each row corresponds to the basis functions derived from $\mathbf{f}_1(x)$, $\mathbf{f}_2(x)$, and $\mathbf{f}_3(x)$, in that order. Notice the DC component present in the second and third rows, and the alternating odd and even harmonics in the corresponding sine and cosine functions.

- The sine function ($\mathbf{f}_1(x) = \sin(x)$) exhibits only odd harmonics (e.g., 1, 3, 5 bumps).

- The cosine function ($\mathbf{f}_3(x) = \cos(x)$) shows even harmonics (e.g., 2, 4, 6 bumps).

- The mixed function ($\mathbf{f}_2(x) = \sin(x) + \cos(x)$) spans both (e.g., 2, 3, 4 bumps).

These frequency patterns can be explained by group theory and the circular harmonics emerging from sine and cosine functions. Specifically, the sine function's odd symmetry leads to odd harmonics, while the cosine function's even symmetry leads to even harmonics.

