# OpenReview forum: "Emergent Geometry in Neural Representations of the Visual World"
_ICLR.cc/2025/Conference — ICLR 2025 Conference Withdrawn Submission_

### Official Review · Reviewer_27v2 · 2024-10-28

**Soundness:** 1
**Presentation:** 3
**Contribution:** 2
**Rating:** 3
**Confidence:** 5

**Summary:**

This paper attempts to test the mirroring hypothesis by probing how the manifold of data is processed by digital twins of primate V4 area and deep neural networks. The results show that rotated images of individual objects form a manifold that persists in neural spaces of digital twin, trained and untrained neural spaces. However, the rotating texture classes with no circular structure in pixel space, evoke an emergent structure only in trained neural spaces. Based on these results, the paper argues for the mirroring hypothesis as a universal property of intelligent systems.

**Strengths:**

The motivation for this work is valuable for both ML and Neuroscience. Studying the universal properties of data processing in natural and artificial intelligence, backed by theory, is crucial for understanding the essentials of intelligence.

The text was written clearly with ample examples from different fields.

**Weaknesses:**

The main weakness can be summarized as not enough evidence to back the claims. Claims related to the universal properties of processing of the data in intelligent systems (natural and artificial) but both experiments (on oriented objects and oriented textures) were performed on a limited set of conditions, and only a specific layer of a specific architecture was used during experiments. Also, it's not clear if the results are specific to robust (sensitive to low-frequencies, re: texture dataset) vs standard (sensitive to high-frequencies re: object dataset) resnet50.

One could think of an alternative explanation:

** In object set (Experiment 1), the variance primarily exists in high-frequency parts of the images, that's why it persists in the early layers of an untrained resnet50. In texture set (Experiment 2), the variance exists in low-frequency parts of the images, that's why a robust resnet50 picks up on that, it's known that robust nets are sensitive to low-frequencies.**

### experiments are insufficient

So here are a few experiments to strengthen the claims ""

1- Claim on 'structured data' preserved even by untrained network representation: only 'layer 3' of untrained resnet50 was tested. What about deeper layers? What about an untrained resnet110? I predict this structure may not survive more aggressive accumulation of non-linearities.
Also, It seems like that as a control to digital twin, standard resnet50 was used in experiment 1 and robust resnet50 was used in experiment 2. Why? How does robust resent50 change the manifold where the variance is mostly in high-frequency parts of the image (Object set, experiment 1.)
To what extent the result depend on this specific image set? How does the manifold structure change without background or with random distinct backgrounds?
> FINDING 1: THE GEOMETRY OF THE DATA MANIFOLD PROPAGATES TO NEURAL SPACE
Does the geometry also preserved in standard resnet50 or deeper untrained neural network, or deeper layers of robust neural network?

2- Claim on the structure 'unstructured data' is recovered by a trained model is only partially tested, as it is unclear what the manifold would look like in a *standard resent50* which was used in experiment 1, or is the recovery of the manifold only a product of adversarial trained neural networks (or networks trained directly on neural data, aka digital twins which many other works already shown direct correspondence between the two). Also, I wonder if really a massive network trained on massive data is needed to extract the manifold in these simple textures, these simple textures could be processed by a low-freq filter (perhaps with an addition of simple non-linearities). One simple test could be to take the model of early vision proposed in Henaff, Goris & Simoncelli 2019 (code available on github) and run the texture images through it to see if that simple network is enough to capture the hidden manifold in this data.
So,
>FINDING 2: THE VISUAL CORTEX RECOVERS THE WORLD MANIFOLD FROM AN
>UNSTRUCTURED DATA MANIFOLD

is really "the network trained on neural data (not visual cortex, because these images were never shown to the animals) recovers the low-dimentional manifold present in the data". What makes a manifold *World manifold*?

3- Experiment 3 needs to test at least standard resnet50 as well, and on more layers of both types of resnet50 before concluding this generalization "is a universal property of intelligent vision systems".

Recommendation
While the work shows promise and moves in an interesting direction, it requires additional control experiments to support its broad claims. Specifically:

Investigation of simpler alternative explanations

Testing across multiple network depths and architectures

Consistent comparison between standard and robust networks

Control image sets (no background or random background for object set)






In summary, although this work seems interesting and in the right direction, at this point it needs more control experiments (see above for a few very simple suggested experiments) or refinement of the claims. At this current format, unfortunately, I can't recommend it for publication. However, I am happy to increase the score if some concerns were addressed.

**Questions:**

- In intro, at least 3 paragraphs were dedicated to explain Digital Twins, however as stated this paper only uses 'unlimited experimentation' ability, with **very** limited experiments actually reported.

- In page 6, "ImageNet-Trained Core:" refers to robust or non-robust ResNet50? Later, both are used in separate experiments, and both were referred to as the control to Digital Twin but here this distinction, (which is important as stated above) was not made clear, why?

- Is there a reason to choose that specific pattern of background for rendering the 3D objects in Figure 3? If so, what is that and if not, do the results change with changing background, or simply 'no background'?

- Is there a reason to call the experiments, **Finding x**  and not **Experiment x** ?

- The first result figure appears on page 7 which shows a long introduction/related work. I wonder if the paper reads better with less irrelevant intro but more relevant experiment and most important parts of the appendix.

---

### Official Review · Reviewer_wq6j · 2024-11-02

**Soundness:** 3
**Presentation:** 3
**Contribution:** 2
**Rating:** 6
**Confidence:** 4

**Summary:**

This paper studied how the low-dimensional structure of representations in convolutional networks reflects the latent structures in the image space. Specifically, it studied in-plane rotation. This paper finds that in-plane rotation of the objects causes the input image space to have a ring-like geometry after dimensionality reduction. Meanwhile, this structure is preserved in the feature space in convolutional networks whether they are trained or not. The in-plane rotation of abstract textures does not have a consistent structure in the image space, but the ring-like geometry can be recovered in networks either trained to predict neural data or on ImageNet. Randomly initialized networks cannot recover this ring-like geometry from texture input. This paper also showed that an orientation decoder trained on one texture can generalize to other textures, while this generalization is impaired when the decoder is trained on features from a random network. Taken together, the author suggests that these results support the mirroring hypothesis of neural representations, that is, the structure of neural representation reflects the latent structure of the world.

**Strengths:**

This paper proposes an interesting framework for analyzing the computational problems of recovering the latent structures in the world. It distinguished between simple processing (the latent structure is readily available in the input space, and the neural representation simply reflects this structure) and intelligent processing (the latent structure is readily available in the input space, and the neural representation recovers this structure). The author then showed two examples. The in-plane rotation of realistic objects corresponds to simple processing, and the in-plane rotation of textures corresponds to intelligent processing. In the first case, either trained or un-trained models can represent the latent structure, while in the second case, only trained models recovered the structure. These methods and results are sound and suggest that training a model either to predict neural activities in V4 or on classification tasks can recover the latent rotation structure in data. This likely exists in real neural data as well.

**Weaknesses:**

My first concern is about the novelty of the findings presented in this paper. It is well-known that networks can learn representations of latent factors of variation in the world, such as rotation (Higgins, et al., 2017, Jahanian, et al., 2019). The in-plain rotation equivariance of CNN representation is studied in depth in the work of Lenc et al., 2015, where they find the representation in AlexNet has significant rotation equivariance, and one can utilize this property to better classify images at different rotations and identify the rotation of objects in novel images. While Lenc et al. did not visualize the data using PCA, and there are differences in the specific methods being used, conceptually, there are many overlaps between the results presented here and Lenc et al. It would be helpful if the author could elaborate more on how their results differentiate from these previous results. If the contribution is mainly in showing the rotation equivariance also exists in the representation of a CNN trained to predict neural data, and thus, it is likely that rotation equivariance also exists in the brain, it would be helpful to show how well the "digital twins" model performs at capturing the neural data (and generalize to the images being tested here).

My second concern is about the scope of the claims. The work frames the "mirroring hypothesis" as a general principle that may apply to many latent variables in the world. For example, in the conclusion section, the author concluded, "We demonstrate that neural representations in area V4 of the primate visual cortex, as well as in trained artificial neural networks, reflect geometric structures of latent variables in the visual world.". However, the evidence provided in this paper is only restricted to 2D in-plain rotation. It is unknown whether the claim holds for many other latent variables, such as translation, 3D rotation, object color, weight, etc. It would be helpful to provide more evidence about other latent variables if this paper makes the claim more general. Or the author could restrict the claim to about in-plain rotation only. In addition, to make a general claim about the mirroring hypothesis, it would be helpful to clarify how to define the "geometric structure of the latent world," which is only vaguely defined here in this paper.

References:
1. Higgins, I., Matthey, L., Pal, A., Burgess, C.P., Glorot, X., Botvinick, M.M., Mohamed, S. and Lerchner, A., 2017. beta-vae: Learning basic visual concepts with a constrained variational framework. ICLR (Poster), 3.
2. Jahanian, A., Chai, L. and Isola, P., 2019. On the" steerability" of generative adversarial networks. arXiv preprint arXiv:1907.07171.
3. Lenc, K. and Vedaldi, A., 2015. Understanding image representations by measuring their equivariance and equivalence. In Proceedings of the IEEE conference on computer vision and pattern recognition (pp. 991-999).

**Questions:**

1. How do the results presented in this paper connect to and differentiate from previous research, especially Lenc et al., 2015?
2. Does the general claim that "neural representation mirrors the geometric structure of the latent world" hold for image variables other than in-plane rotation, such as translation, 3D rotation, object color, weight, etc., or is it just specifically about in-plane 2D rotation?
3. How to clearly define the "geometric structure of the latent world"?

Minor points:
1. figure 3, c, does not have a middle panel
2. it is unclear what does DDM mean in section 4.4
3. in section 4, is the robust ResNet model the same as the ImageNet-Trained Core model mentioned previously?
4. in the conclusion section, The claim that "We demonstrate that neural representations in area V4 of the primate visual cortex ... reflect geometric structures of latent variables in the visual world." lacks evidence since it is unclear how the CNN trained to predict neural data perform and whether they still capture neural data well in the images tested in this study.

---

### Official Review · Reviewer_ny1S · 2024-11-04

**Soundness:** 2
**Presentation:** 3
**Contribution:** 1
**Rating:** 3
**Confidence:** 5

**Summary:**

This paper frames visual processing as the ability of neural population representations to recover world structure from input data structure. The authors fit a deep neural network (DNN) to macaque V4 responses and analyze how V4 responses successfully recover information such as orientation and position, regardless of whether the input data reflects the world structure. This type of representation, demonstrated in both brain-trained and image-trained models, has been shown to facilitate zero-shot generalization.

**Strengths:**

1.	Despite some technical details, the overall logic of the paper is well-written and clear.  I can understand the authors very well.
2.	This study used monkey physiological data for model training

**Weaknesses:**

1.	I doubt the very essential novelty for this study.
2.	The abstract is obscure and contain little information. There is no real meat there.
3.	The descriptions of model training details are unclear.

**Questions:**

1.	My main concern with this paper is that the study dedicates an entire article to addressing very basic and somewhat trivial questions. While I agree that we should distinguish world space, data space, and neural space in visual processing, and I agree with the focus on how neural space can recover world space variables from data space inputs, this represents the most fundamental function of both the brain and artificial neural networks (ANNs). If the brain or an ANN cannot perform this task, it essentially indicates failure in function. For instance, if an ANN fails to recognize a face across different lighting and noise conditions, then it has failed in face recognition. Conversely, if the ANN is trained to perform this task effectively, it stands to reason that two faces perceived as similar should also be represented similarly at the neural level. If not, what would be the alternative?

2.	I’m not entirely clear on the mirror hypothesis. For me, this is basically the same as representational similarity (Kriegeskorte&Wei, Nat Rev Neurosci, 2012). In the biological brain, if two stimuli are perceived as similar, then their neural representations are generally similar—this principle has been established in numerous studies using representational similarity analysis. So, doesn’t the mirror hypothesis essentially reiterate what is already widely known and addressed by representational analysis? The idea that "similar-looking stimuli yield similar neural representations" has been well-validated, and we don’t really need manifold theory to explain it.

3.	Additionally, the paper discusses structured and unstructured information in data space. When world structure is preserved in data—such as with simple properties like orientation or shape—we already know that population representations will be continuous. Even when world structure isn’t maintained in the data, as long as the brain perceives it coherently, the neural representation must still be continuous. For example, a person’s face can be perceived under varying lighting conditions or occlusions; while the data (image) space may lack consistent structure, the brain can still extract identity. Achieving this is a basic function of any effective visual system. Is this really new?

4.	In Section 4.1, what are the differences between the "Digital Twin – Untrained" and "Random Core" conditions? It seems that both have the same architecture with randomly initialized weights.

5.	Regarding Figure 3, when analyzing the neural manifold, are you using data from ResNet Layer 3, or are you directly using the 1,244 neurons? It’s unclear which part of the digital twin model is being used to calculate the manifold.

---

### Note · Authors · 2024-11-25

**Comment:**

We thank the reviewers for their time and consideration.  We have decided to withdraw in order to have sufficient time to pursue our desired changes to the paper.  We will take into account the reviewers’ thoughtful feedback when implementing further revisions on this work.

**Withdrawal Confirmation:**

I have read and agree with the venue's withdrawal policy on behalf of myself and my co-authors.